# A Flexible Metamaterial Based Printed Antenna for Wearable Biomedical Applications

**DOI:** 10.3390/s21237960

**Published:** 2021-11-29

**Authors:** Ammar Al-Adhami, Ergun Ercelebi

**Affiliations:** Department of Electrical Electronics Engineering, Gaziantep University, Gaziantep 27310, Turkey; ercelebi@gantep.edu.tr

**Keywords:** SAR, flexible antenna, wearable, MTM

## Abstract

This paper presents a microstrip antenna based on metamaterials (MTM). The proposed antenna showed several resonances around the BAN and ISM frequency bands. The antenna showed a suitable gain for short and medium wireless communication systems of about 1 dBi, 1.24 dBi, 1.48 dBi, 2.05 dBi, and 4.11 dBi at 403 MHz, 433 MH, 611 Mz, 912 MHz, and 2.45 GHz, respectively. The antenna was printed using silver nanoparticle ink on a polymer substrate. The antenna size was reduced to 20 × 10 mm^2^ to suit the different miniaturized wireless biomedical devices. The fabricated prototype was tested experimentally on the human body. The main novelty with this design is its ability to suppress the surface wave from the patch edges, significantly reducing the back radiation toward the human body when used close to it. The antenna was located on the human head to specify the specific absorption rate (SAR). It was found in all cases that the proposed antenna showed low SAR effects on the human body.

## 1. Introduction

In the last decade, electronic systems miniaturization has led to an increased demand for wearable devices that can monitor the human body’s functions [1]; thus, wearable health management systems became a most attractive field for researchers. This is due to the fact that such wearable devices can function well enough to replace several medical instruments when embedded in smart clothes [2]. In order apply such technology, researchers developed a variety of miniaturized antennas with adequate performance [3]. However, to employ wearable systems in biomedical applications, several practical specifications in antenna design must be considered, such as small size, low weight, power consumption, and flexible structure [4]. For this, a number of microstrip antennas were introduced as one of the most desired categories for wearable applications when mounted close to the human body [5].

Subsequently, significant demands for wearable antennas to be compact and to not have unnecessary setup requirements were increased for self-adaptive wireless systems [6]. It is worth mentioning that one of the main research interests of wearable antennas is wireless body area network applications [7]. Consequently, printed circuit antennas became attractive due to their unique properties, such as their being very compatible with the requirements of wearable systems [8]. The main important advantages of using printed antennas for wearable systems are their cost-effectiveness, design simplicity, and their relative biocompatibility [9]. Moreover, one of the important advantages of using printed circuit antennas with wearable systems is that they can be mounted on flexible and/or semi-flexible substrates, maintaining their performance against physical bending and twisting effects [10]. Thus, attention must be paid during the design process to consider employing suitable materials for the wearable antenna fabrication [11]. Such considerations have a direct effect on a fabricated antenna’s efficiency and bandwidth, where they are usually highly affected by substrate thickness and the dielectric constant [12].

Currently, significant interest has been raised with regard to flexible and semi-flexible substrates for telemedicine and biomedical applications [13]. Much attention was paid to the use of flexible and semi-flexible materials for wearable nodes production [13]. With the great advancements in wearable antenna technologies, a significant development has been observed in their utilization for implantable devices [14]. However, operating antennas at low frequency ranges with miniaturized size for wearable devices without affecting radiation efficiency and gain is one of the more relevant challenges in the current state of the art [15]. To resolve such a problem, various efforts have been made to reduce the size of wearable antennas using reactive loads [16], utilizing materials with high permittivity [17], and using vias [18], shorting posts [19], and fractal geometries to increase the current path [20]. When the mentioned techniques have been applied, the antenna size has been significantly reduced. However, different difficulties appeared, such as bandwidth reduction, design complexity, and gain degradations [8]. In addition, even though wearable antennas were applied in different technologies based on smart clothes, they still suffered from different issues, such as their influences on the hosting body, the effect of their location on different body parts on their operation, and the possible occurrence of significant efficiency and reliability losses due to antenna deformation [18]. Therefore, a critical issue in wearable antenna design that must be considered is minimizing electromagnetic leakages toward the human body.

In another aspect, biocompatible materials such as polyamide substrates have recently attracted researchers to include them in their designs for wearable antennas [15]. This is due to their excellent mechanical properties, including but not limited to their light weight, flexibility, tensile strength, durability, high-heat resistance (up to 400 °C), excellent electrical properties, high moisture release characteristics, and low moisture uptake [17]. On the other hand, due to the properties of human tissue that have high permittivity, when the human body is exposed to the electromagnetic waves that radiate from a wearable antenna, the body absorbs a large amount of this energy [19]. As is well known, the parameters of a wearable antenna are decreased noticeably during its functioning near the human body, which causes significant problems in the wireless communication field [5]. Furthermore, electromagnetic waves absorbed by the human body have undesirable environmental and biological impact [6]. Several types of wearable antennas have been developed in the past. The authors of [7] designed a compact triangular patch antenna; however, the antenna has a very narrow operating bandwidth. A low-sized cpw-fed slot antenna with floating ground plane for ISM band was designed in [7]; however, the designed antenna had a very low fractional bandwidth (6% at the center frequency of 5.825 GHz). Several other wearable antennas such as electromagnetic bandgap (EBG)-based antennas [2] and substrate-integrated waveguide (SIW)-based antennas [10] have been designed; however, all of these antennas have had a narrow bandwidth.

In this paper, a low profile and flexible printed wearable antenna was designed for ISM bands including 403 MHz, 433 MH, 611 Mz, 912 MHz, and 2.45 GHz for remote health-monitoring applications. The antenna was based on a polyimide substrate, which is known for its flexibility and robustness. The performance of the proposed antenna was acceptable in terms of gain, operating bandwidth, and efficiency in the bending scenarios. In addition, the antenna has appropriate gain, acceptable bandwidth, and high efficiency in on-body worn scenarios. Moreover, the antenna has a reasonably low SAR value when mounted on the human body.

## 2. Antenna Geometry

The antenna is constructed from an MTM patch of a complementary Minkowski fractal geometry to realize multiple frequency resonance within a miniaturized size. The patch is fed with 50 Ω coplanar waveguide (CPW) to achieve excellent matching at several bands [6]. Two matching circuits are introduced (see Figure 1a) between the patch structure and the CPW ground plane to reduce the reflection effects [10]. Nevertheless, the advantage of adding those matching circuits is to suppress the surface waves along the patch edges [12].

In the same context, reducing the effects of the surface waves reflections from the substrate edges is achieved by etching MTM defects from the proposed CPW ground plane [18], which minimizes the back radiation. In turn, the electromagnetic leakage toward the human body would be reduced significantly [10]. As seen in Figure 1b, the antenna dimensions are 20 mm × 20 mm when printed on a polymer substrate of 0.3 mm thickness with relative permittivity of 3.5 and loss tangent of 0.0001. The antenna back panel is covered with a partial ground plane to reduce the effects of the capacitive coupling with the patch structure [13].

Finally, the antenna structure is printed using conductive ink based on silver nanoparticles of conductivity of around 1.3 × 10^6^ s/m. The novelty of the proposed design is that by considering the complementary Minkowski geometry, such a structure can therefore dominate the magnetic field instead of the electrical field [10]. Consequently, SAR effects would be insignificant on human tissue where the human body permeability is unity [13].

## 3. MTM Characterizations

The proposed MTM is defined as sub-wavelength composites of right–left hand structures with negative ε_r_ and μ_r_ at the frequency bands of interest. The resonance frequency of the proposed MTM unit cell is affected by substrate height and permittivity [12]. Therefore, CSTMWS based on the finite integral technique (FIT) [21] was invoked to investigate the proposed MTM behavior in terms of S-parameters and dispersion diagram. As seen in Figure 2, the proposed unit cell is etched from a transmission line ground plane to obtain quasi-TEM-like modes [9]. In that simulation, the upper and lower ±y axes faces are considered perfect electrical conductors (PEC). The other two faces along the ±x axes are subjected as perfect magnetic conductors (PMC). The waveguide ports are assigned ±z axes. It is important to mention that the proposed unit cell is normal to the excitation ports. This was taken into consideration because the proposed unit cell is etched from the patch surface in which the electric field would be normal to the patch surface [7].

Here, the proposed unit cell structure is characterized in terms of S-parameters, S_11_ and S12, and dispersion diagram. As seen in Figure 3a, the proposed unit cell shows more than single frequency resonance within the bandwidth of interest. The authors treated this unit cell as explained in [9]. Therefore, based on the resulting variation in the wave velocity for such unit cell at the first Brillion zone, see the dispersion diagram in Figure 3a, the wave propagation band gap can be evaluated. From the observed results at the first transverse electrical and magnetic modes, the proposed unit cell shows a band gap between the frequency range of 1 GHz and 2.43 GHz; which make it a good candidate for such applications [12].

Based on the simulation results, the proposed unit cell equivalent circuit model is derived as seen in Figure 4. The equivalent capacitances of coupling between the unit cell and adjacent cells are indicated as Left Hand (LH) capacitor of (C_LH_), and the fractal slot is considered as an inductor (L_LH_) where the magnetic current motion in the air traces can be magnified [12]. This inductor is equivalent to the magnetic field that is stored inside the rings’ fractal slots [6]. The other presented elements are considered for the traditional medium of propagation due to the Right Hand (RH) transmission part, which is given by L_RH_, R_RH_, G_RH_, and C_RH_ as inductance, resistance, conductance, and capacitance elements. These parameters are driven by coupling effects between the T-resonator the fractal shape in the unit cell [12].

The lumped elements of the proposed circuit in Figure 3 were evaluated based on a parametric study inside Advanced Design System (ADS) environments, and they are listed in Table 1. Therefore, the frequency resonance can be tuned by varying the gap width between the T-resonator in the unit cell and the fractal geometry. Consequently, a parametric study was applied by changing the gap between the T-resonator, and the fractal geometry (G) was conducted to determine the operating resonance frequency, which led to the desired unit cell properties at the frequency band of interest. Figure 5 displays the conducted parametric study on the parameter G from 0.1 mm to 0.7 mm, with a step of 0.1 mm. The achieved results in terms of S_11_ and S_12_ spectra were monitored. It was observed that |S_21_| spectra were significantly affected by g1 value variation. Therefore, the resonant frequency shifted from 0.38 GHz at g1 = 0.3 mm to 0.68 GHz at g1 = 0.7 mm. It was observed from the obtained results at g1 = 0.5 mm that the matching impedance was significantly enhanced at 0.4 GHz.

## 4. Design Methodology

In this section, a parametric study based on numerical simulations was conducted inside CSTMWS environments based on the finite integral technique algorithm [21]. For this study, the antenna geometrical parameters were changed parametrically to realize the optimal antenna performance in terms of bandwidth and gain. Therefore, the antenna S_11_ spectra, gain spectra, and radiation patterns were monitored by varying the antenna dimensions. The proposed antenna dimensions were swept to maintain the frequency bands around 403 MHz, 433 MH, 611 Mz, 912 MHz, and 2.45 GHz. For this, the parametric study was conducted by changing the antenna design parameters as follows:

### 4.1. Patch Design

A parametric study was applied to the proposed patch by changing the geometry from solid patch to defected MTM patch based on Minkowski fractal. As seen in Figure 6a, the S_11_ spectra of the proposed patch showed more frequency modes than the identical ones based on the solid patch. This is due to the fact of the effects of the fractal geometry on the generated modes because of the multi path current motion [3]. Moreover, the proposed antenna gain was enhanced significantly in comparison with the sold patch, as seen in Figure 6b. For example, the antenna gain at 403 MHz was found at about 1 dBi, and at 2.45 GHz it was enhanced to 4.1 dBi. Such enhancements are attributed to surface wave suppression [9].

### 4.2. Matching Circuit

The authors investigated the effect of changing the matching circuit length on the antenna performance in terms of the S_11_ spectrum only. Therefore, three lengths were examined, starting from 4 mm to 12 mm in a step of 4 mm. Figure 7 shows the impact of changing the matching load length (L) on the evaluated S_11_ spectra. The obtained results show that changing the mating length directly affected the antenna bandwidth due to the fact of tuning the real part of antenna impedance with respect to the characteristic impedance of the source [7]. Nevertheless, with increasing L, a significant reduction in the field fringing from substrate edge [9] consequently enhanced the antenna bandwidth. Therefore, the authors decided to fix the matching circuit length to 12 mm.

### 4.3. MTM Effects

In this section, the authors decided to conduct a parametric study on the effects of the MTM row number, see Figure 1a, on the proposed antenna performance in terms of S_11_ and gain spectra. The results of this study are shown in Figure 8; the results again reveal excellent enhancements in the antenna bandwidth by increasing the MTM array from 1 row to 3 rows with a step of 1 row. This is attributed to the fact that suppressing the surface wave was increased by increasing the number of MTM rows [9].

## 5. Bending Effects and Radiation Leakages

In this section, the variation in the antenna S_11_ spectra was monitored after bending the antenna close to the human head. Additionally, SAR effects are discussed with respect to human tissue. This study was conducted by invoking the voxel model of Sam phantom structure inside CSTMWS environments as follows:

### 5.1. Bending Effect

The antenna was subjected to the bending effects on a cylindrical geometry with different angles from 0° to 45° with a step of 15°. It was observed that when the antenna was subjected to the bending effects it showed insignificant changes in terms of S_11_ spectra, as seen in Figure 9; this was due to the current motion on the fractal patch structure not being significantly affected because of the MTM structure at the feed point [8].

### 5.2. Radiation Leakage

Since the proposed study was subjected to the applications of wearable applications, a SAR study based on radiation leakage from the antenna toward the human tissue was studied. The effects of the absorbed radiation in the human head are studied numerically in this section. In the simulation, the input power level was considered to be 1 mW. We focused on the proposed antenna design to realize minimum leakage toward human tissue by adding two fractal unit cells near the feeding port. In Figure 10a, the values of the radiation leakage in terms of electric field (E-Field) strength are presented. In this study, the antenna was mounted close to the human head with varying distances from 0 mm to 50 mm, with a step of 5 mm. It is important to mention that the bending effect was applied, here at 45°, without considering the flat case in the comparison. The quantity of absorption was given by evaluating the SAR results on the SAM model of the human body inside CSTMWS environments, as in Figure 10b. It is important to mention that the SAM model was simulated inside CSTMWS environments with a resolution of 1 mm^3^ [13]. As seen in Figure 10b, the SAR quantity was found to be 0.25 W/kg and 0.33 W/kg at 403 MHz and 2.45 GHz, respectively, with a distance of 5 mm. Finally, the field strength leakages from the antenna toward the human head were about 101 mV/m and 133 mV/m at 403 MHz and 2.45 GHz, respectively.

## 6. Experimental Results and Discussions

After arriving at the optimal antenna design, the authors decided to fabricate the antenna design as presented in Figure 11. The proposed antenna was fabricated using a conductive ink of silver nanoparticle printed with a Fujifilm Dimatix materials printer. The fabricated antenna was tested experimentally including: S_11_ spectrum, radiation patterns, and field radiation leakages.

During the measurement process, the authors invoked the use of an RF chock with 50 coaxial cables connected to a professional network analyzer of the Agilent family of PNA 8720. The antenna measurements were performed inside an RF anechoic chamber as follows:

### 6.1. Antenna Characterizations

The proposed antenna was fabricated and tested. In this matter, S_11_ spectra and radiation patterns of the proposed antenna at different frequency bands with different bending scenarios in the free space were measured. Later, the same measurements were performed again when the antenna was mounted close to the human head. The proposed antenna was measured within the frequency band from 0.1 GHz up to 3 GHz. As seen in Figure 12, the antenna S_11_ spectra were measured for bended profile, at 15° and flat case, in which it was mounted off and on the human head. In Figure 12, the antenna S_11_ spectrum is presented in the free space based on the flat case; it was found that the proposed antenna showed a frequency resonance at 403 MHz, 433 MH, 611 Mz, 912 MHz, and 2.45 GHz, with an S_11_ value below −10 dB. Subsequently, when the antenna was subjected to bending at 15°, the antenna S_11_ and the frequency resonance values were not significantly affected, as seen in Figure 12b, which agrees with the results in the previous section. After that, the antenna based on the flat profile was mounted close to the human head to assess whether there were human tissue effects from the antenna S_11_ spectra, as depicted in Figure 12c. We found that frequency resonance was shifted slightly at high frequency bands, around 2.45 GHz; however, lower frequency bands showed insignificant changes. This was likely due to the fact that wave scattering from the antenna edges at low frequencies were insignificant [10]. Next, when the antenna was subjected to bending effects and placed on the human head, the antenna S_11_ spectrum, generally, was not changed significantly, as presented in Figure 12d.

Next, the antenna gain spectra were measured for the proposed antenna within the frequency band of interest as well as in the same scenarios. It was found that the proposed antenna realized insignificant variation in the antenna gain in comparison with the effects of bending or being mounted on the human body, as seen in Figure 13.

Subsequently, the antenna radiation patterns at 403 MHz, 433 MH, 611 Mz, 912 MHz, and 2.45 GHz were measured for the flat and bended profiles in the free space as seen in Figure 14. It was found that the proposed antenna showed an absolute gain of 1 dBi, 1.24 dBi, 1.48 dBi, 2.05 dBi, and 4.11 dBi at 403 MHz, 433 MH, 611 Mz, 912 MHz, and 2.45 GHz, respectively. Such gain values were found to be very stable for short and medium wireless communication systems. Additionally, the antenna performances were not found to be significantly affected with bending, making it an excellent candidate for wearable application. It is important to mention that during the measurement the input power was considered to be 1 mW. In this section, a comparison between different antennas that were introduced in the literature for modern applications from different aspects are compared to the proposed antenna designs. It is clear from Table 2, that the proposed antenna, to the best of the authors’ knowledge, provides excellent gain with mutable frequency bands with a limited area in comparison with data on those antennas that were published previously.

As a next step, the authors applied the same study when the antenna was mounted close to the human head. It was found that most of the radiation patterns were directed away from the human head due to the effects of the MTM at the feed position, as discussed later. The antenna radiation patterns for both flat and bended profiles close to the human head were insignificantly affected. It is important to mention that the proposed antenna was located at a 5 mm distance from human tissue. In general, the simulated and measured results were in good agreement for all cases, as depicted in Figure 15. The antenna showed good immunity against bending without significant effects of the human head on antenna performance.

In Figure 16, the proposed antenna performance was measured in terms of *S*_11_ and gain spectra in the free space and when mounted on the human head. During the measurements, the antenna was bended from 0° to 45°, with a step of 15°. It was found that the proposed antenna performance was not changed significantly at low frequency bands, showing excellent stability against bending effects. Such stability was achieved due to the surface wave suppression at the patch and substrate edges [20]. Nevertheless, bending effects usually result in a significant change on the frequency resonance of wearable antennas; this is usually attributed to the effects of the equivalent antenna capacitive variation that would increase with the increase in antenna bending effects [2]. However, in our case, the antenna patch was constructed from a fractal antenna structure that was shaped from a meander line within a limited area; such a configuration generates a magnetic field component normal to patch edges that reduces the capacitive effects [17] and maintains the surface current distribution on the patch when it is subjected to bending [18].

### 6.2. Radiation Leakage

The radiation leakage from the proposed antenna toward the human tissue was measured in terms electrical field strength using a field probe meter. As seen in Figure 17, the proposed antenna showed a field strength leakage toward the human head of about 101 mV/m at 403 MHz and 133 mV/m at 2.45 GHz for the flat profile. The radiation leakage was measured at different distances between the proposed antenna on bended profile and the human head of about 5 mm up to 25 mm, with a step of 5 mm, as seen in Figure 17. It was found that the radiation leakage was reduced significantly after 25 mm from the proposed antenna for both flat and bended profiles. These measurements were conducted using a TM-195 RF 3-axie field strength meter.

In Table 3 is presented a comparison between the proposed antenna performance and other published results in the literature in terms of SAR amount at different frequency bands. It was found that the proposed antenna showed less SAR amount than other considered references.

## 7. Conclusions

The proposed antenna was designed based on MTM structure of a composite right–left array. The antenna was fabricated on a polymer flexible substrate with 0.3 mm thickness to suit wearable applications. Therefore, the antenna size was miniaturized to 20 mm × 10 mm to suit different miniaturized wireless portable devices. The proposed antenna was fabricated by printing technology using conductive silver nanoparticle ink. The antenna prototype was tested experimentally to show several resonances at 403 MHz, 433 MH, 611 Mz, 912 MHz, and 2.45 GHz, which are suitable for BAN and ISM bands. The measured antenna gain was found to vary from 1 dBi to 4.11 dBi. The measured antenna performance was applied in two scenarios based on flat and bended profiles when mounted on the human head and away from it. To assess the proposed antenna’s effects on human tissue, the proposed antenna was located close to the human head by measuring the SAR rates and electric field strength leakages. It was found that the proposed antenna showed low SAR effects of about 0.25 W/kg and 0.33 W/kg at 403 MHz and 2.45 GHz, respectively. Finally, the proposed antenna measurements were compared with the simulated results, showing good agreement.

## Figures and Tables

**Figure 1 sensors-21-07960-f001:**
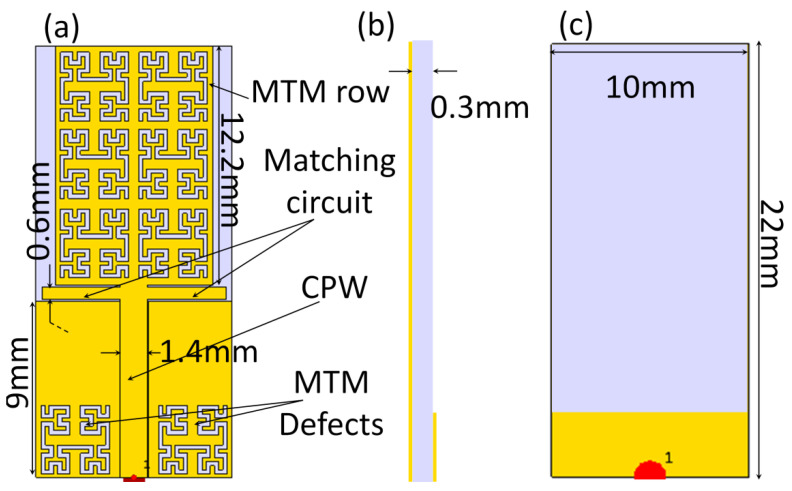
The antenna geometrical: (**a**) front view, (**b**) side view, and (**c**) back view.

**Figure 2 sensors-21-07960-f002:**
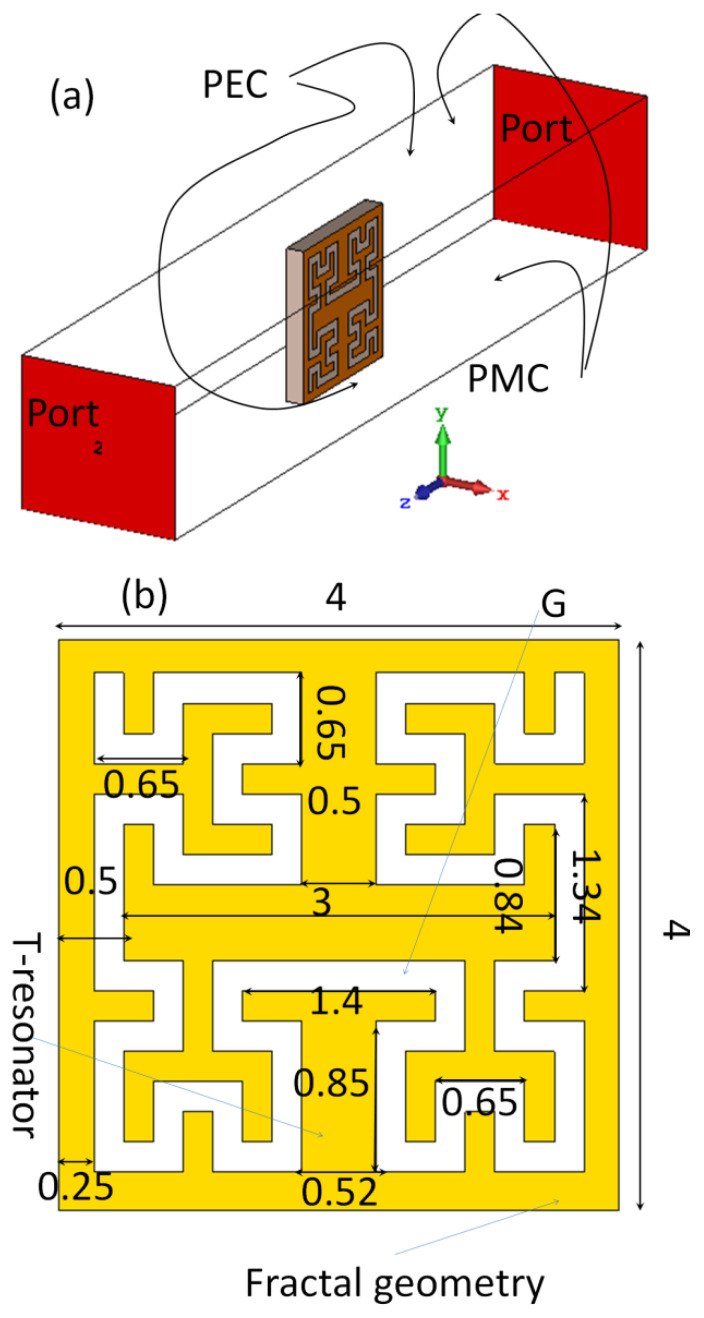
EBG unit cell: (**a**) CSTMWS numerical setup and (**b**) unit cell dimensions in mm scale.

**Figure 3 sensors-21-07960-f003:**
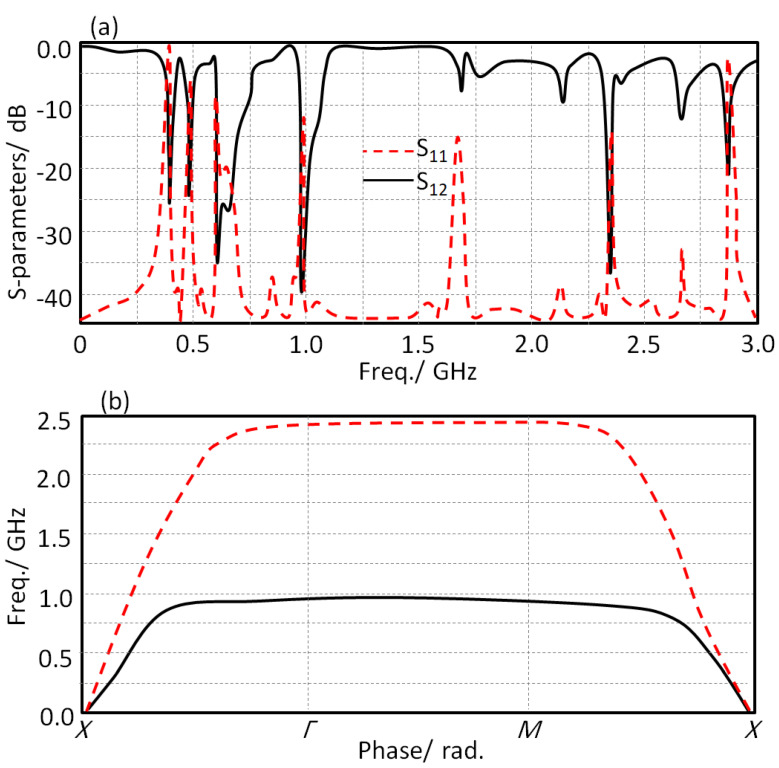
Unit cell characterizations: (**a**) S-parameters and (**b**) dispersion diagram.

**Figure 4 sensors-21-07960-f004:**
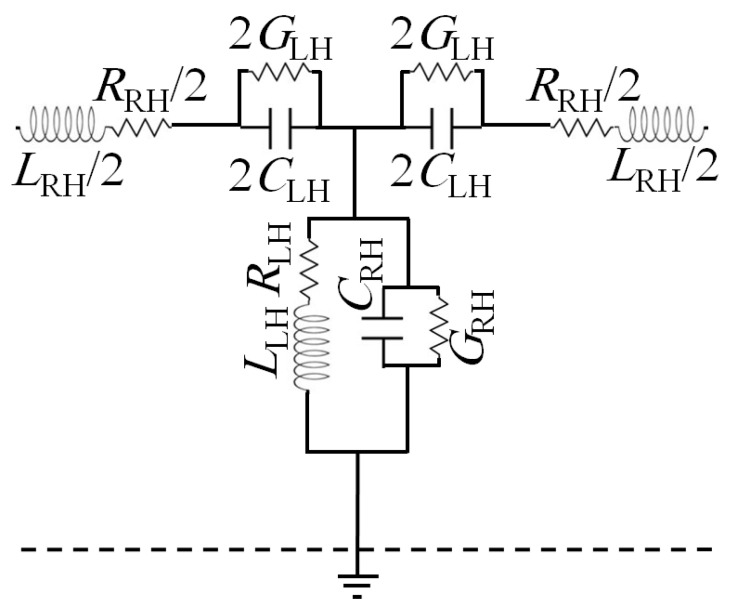
Equivalent circuit of the proposed unit cell.

**Figure 5 sensors-21-07960-f005:**
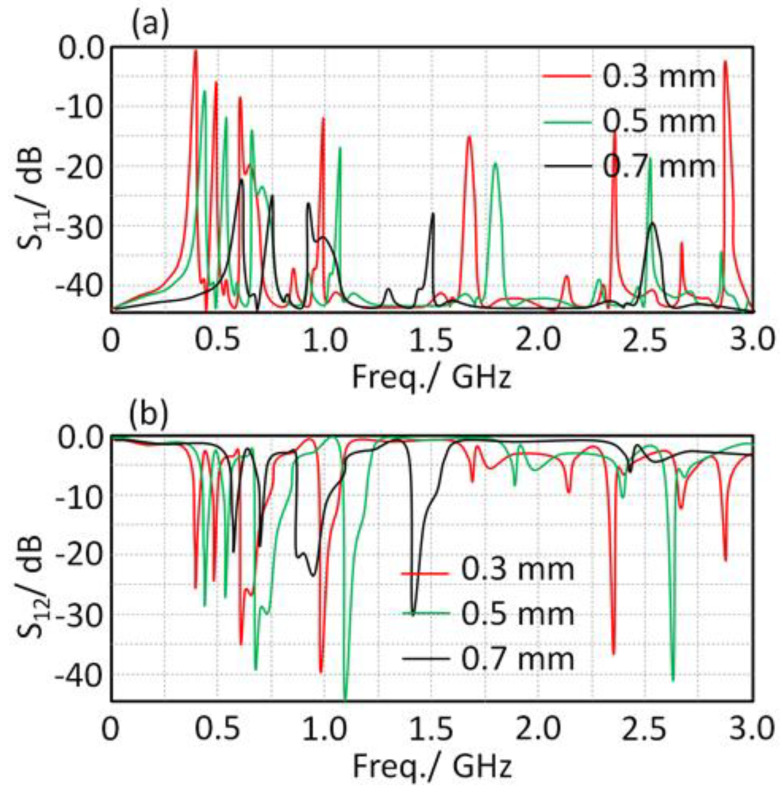
Varying effects of G dimension on the S-parameter spectra: (**a**) S_11_ and (**b**) S_12_.

**Figure 6 sensors-21-07960-f006:**
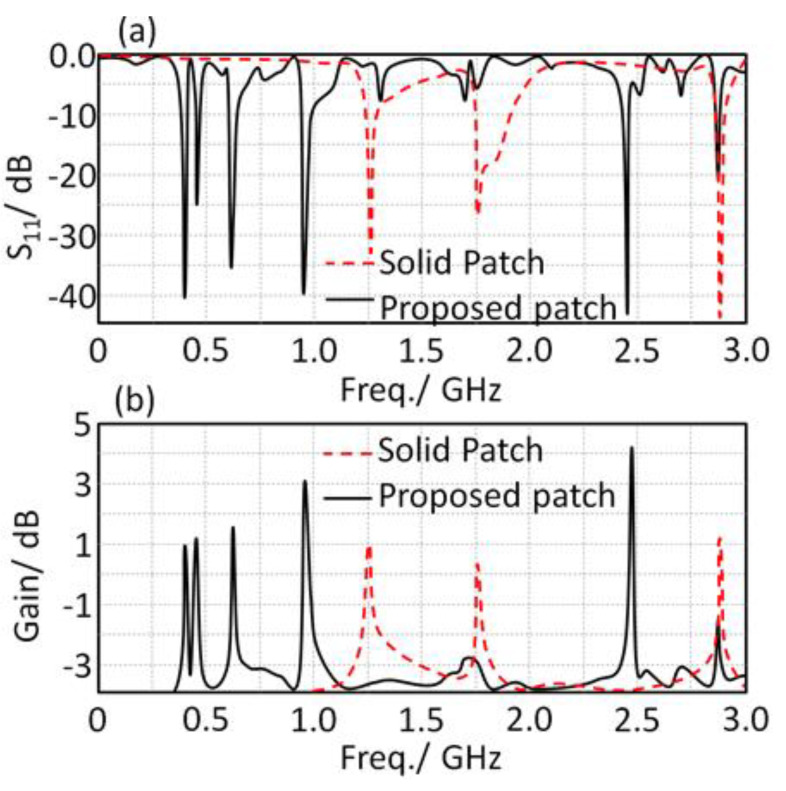
Effects of changing the patch configuration shape: (**a**) S_11_ spectra and (**b**) gain spectra.

**Figure 7 sensors-21-07960-f007:**
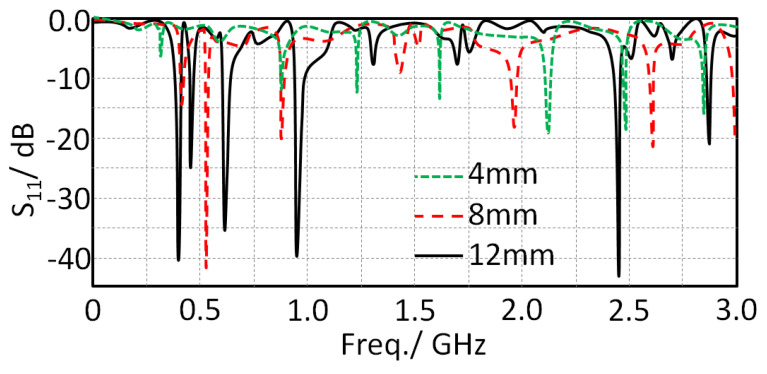
Effects of changing the matching impedance length on S_11_ spectra.

**Figure 8 sensors-21-07960-f008:**
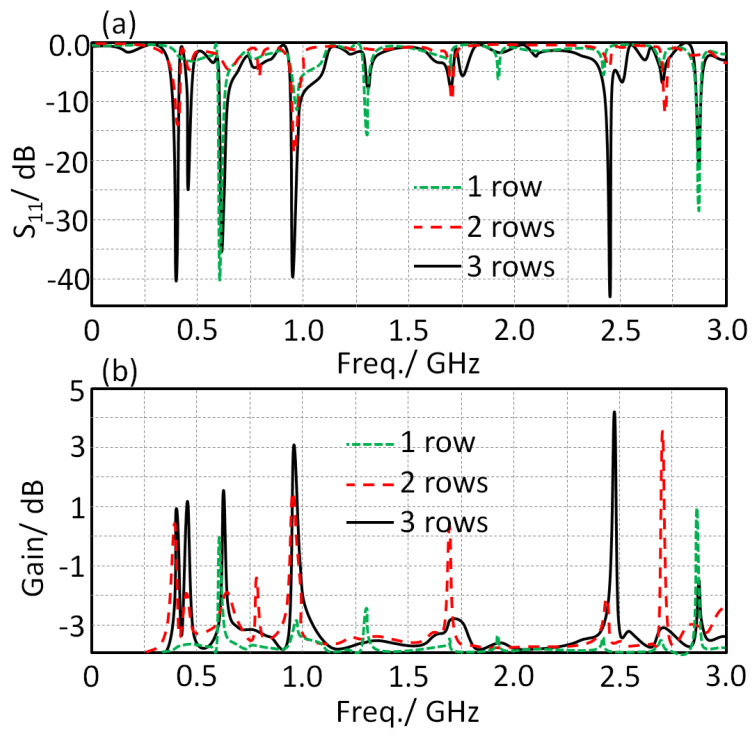
Antenna performance by varying the MTM rows: (**a**) S_11_ and (**b**) gain spectra.

**Figure 9 sensors-21-07960-f009:**
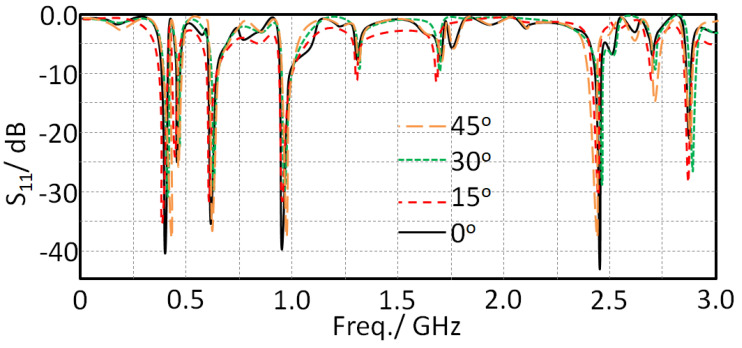
Antenna S_11_ spectra with varying the bending angle.

**Figure 10 sensors-21-07960-f010:**
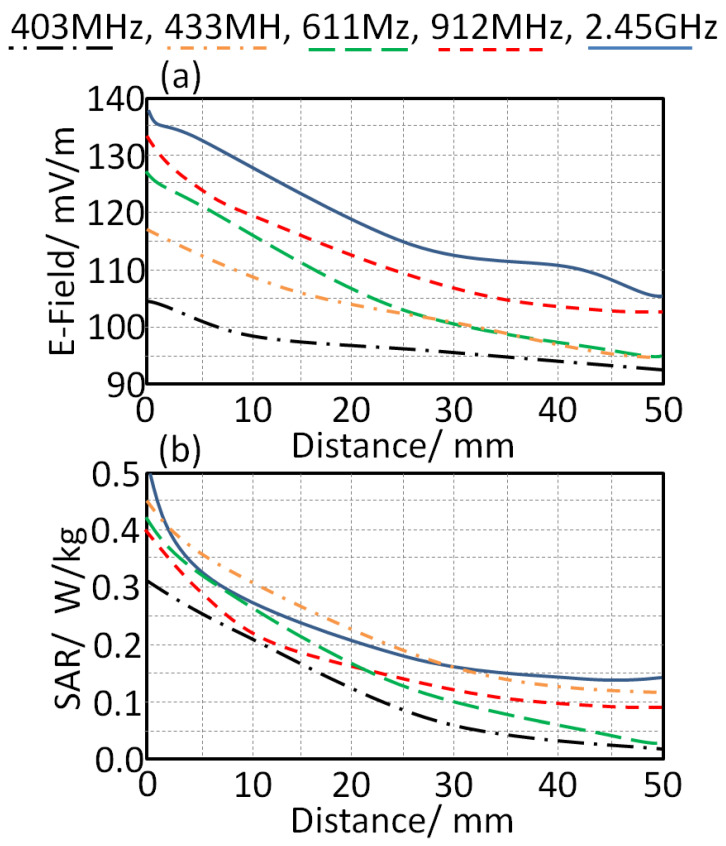
Antenna radiation leakage in terms of: (**a**) E-field and (**b**) SAR.

**Figure 11 sensors-21-07960-f011:**
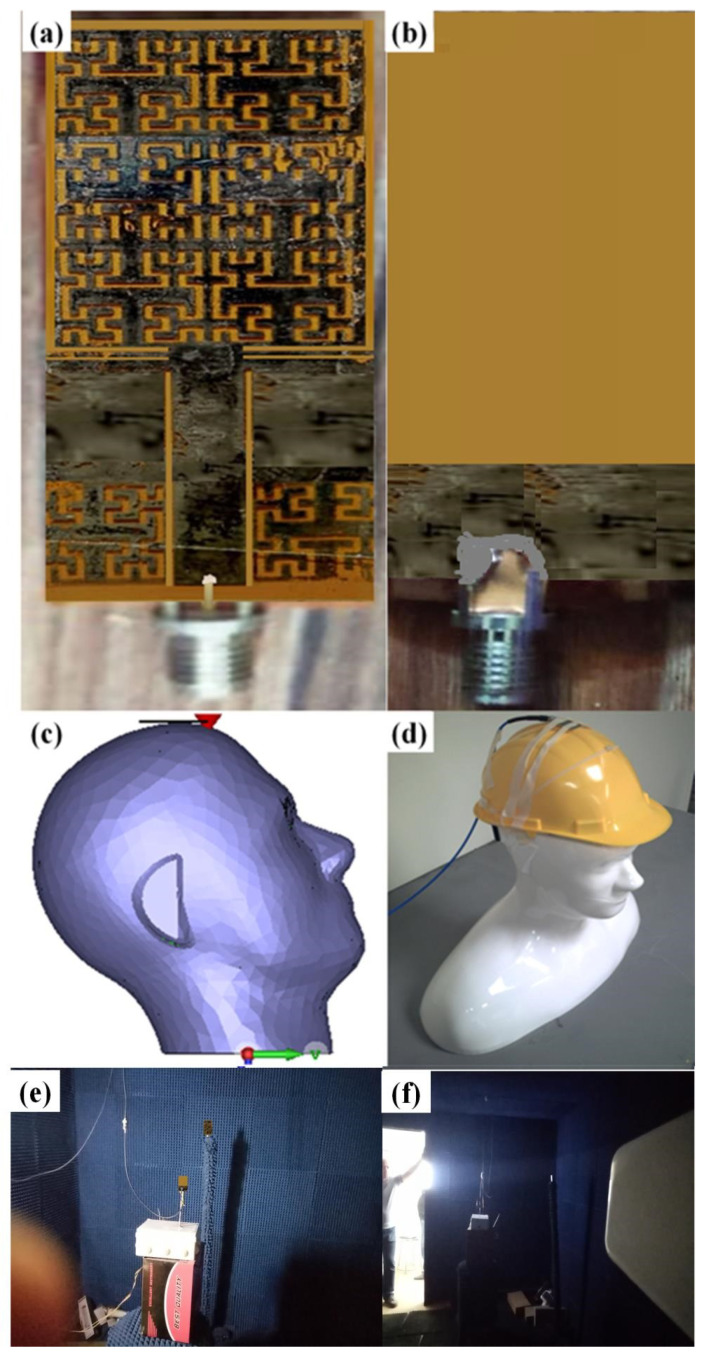
Fabricated antenna prototype: (**a**) front view, (**b**) back view, (**c**) SAM model inside CSTMWS, (**d**) SAM model during the experimental measurements, (**e**) antenna gain measurement, and (**f**) radiation pattern measurement.

**Figure 12 sensors-21-07960-f012:**
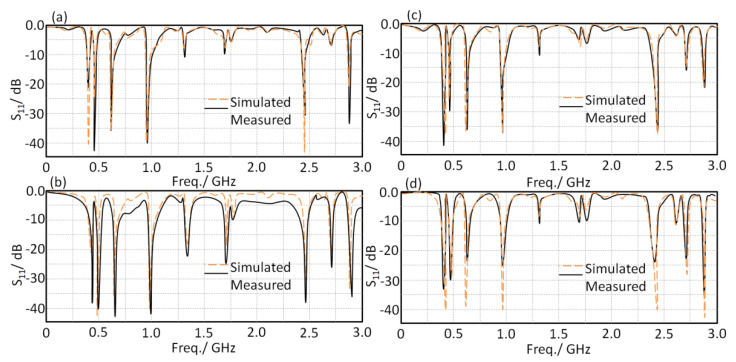
Antenna S_11_ spectra for two proposed profiles: (**a**) flat in free space, (**b**) flat on the human head, (**c**) bended in free space, and (**d**) bended on the human head.

**Figure 13 sensors-21-07960-f013:**
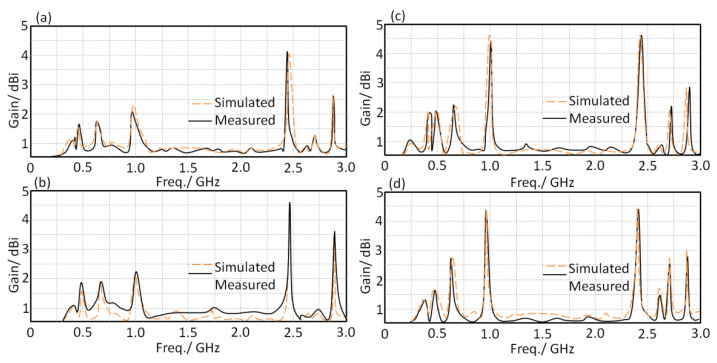
Antenna gain spectra for two proposed profiles: (**a**) flat in free space, (**b**) flat on the human head, (**c**) bended in free space, and (**d**) bended on the human head.

**Figure 14 sensors-21-07960-f014:**
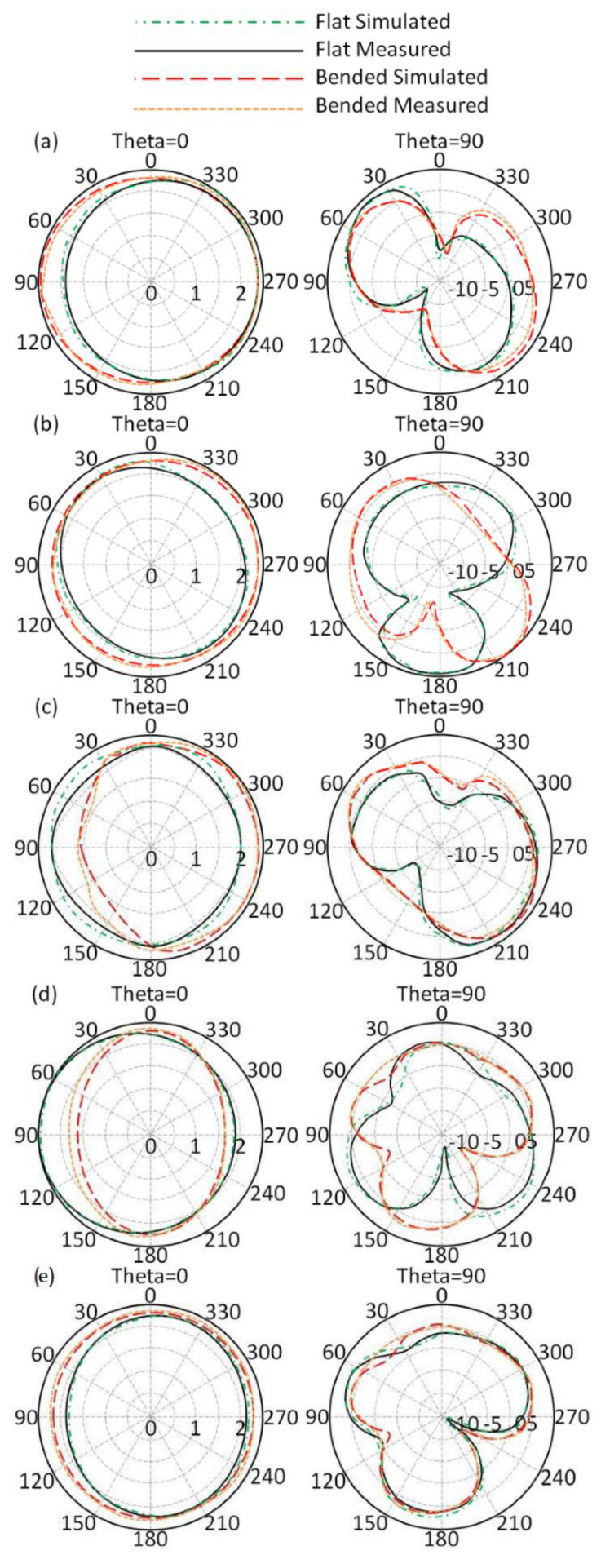
Antenna radiation patterns in free space: (**a**) 403 MHz, (**b**) 433 MH, (**c**) 611 Mz, (**d**) 912 MHz, and (**e**) 2.45 GHz.

**Figure 15 sensors-21-07960-f015:**
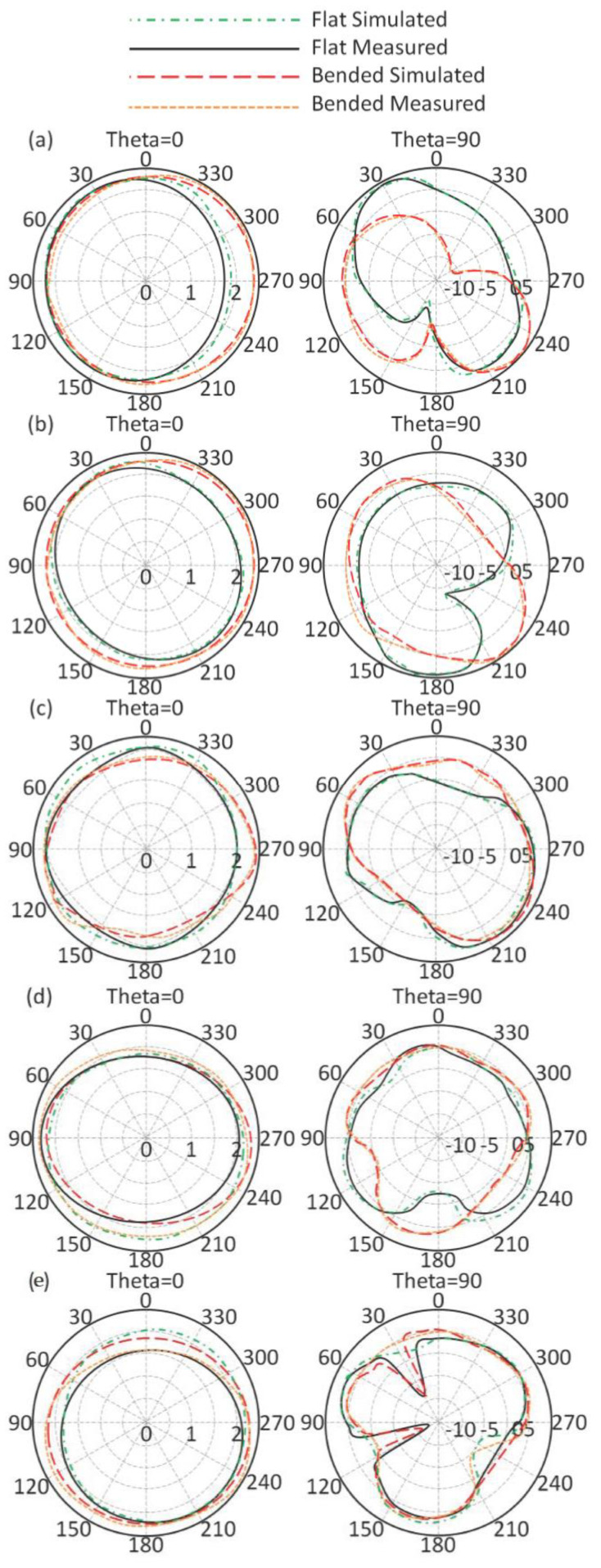
Antenna radiation patterns when close to the human body: (**a**) 403 MHz, (**b**) 433 MH, (**c**) 611 Mz, (**d**) 912 MHz, and (**e**) 2.45 GHz.

**Figure 16 sensors-21-07960-f016:**
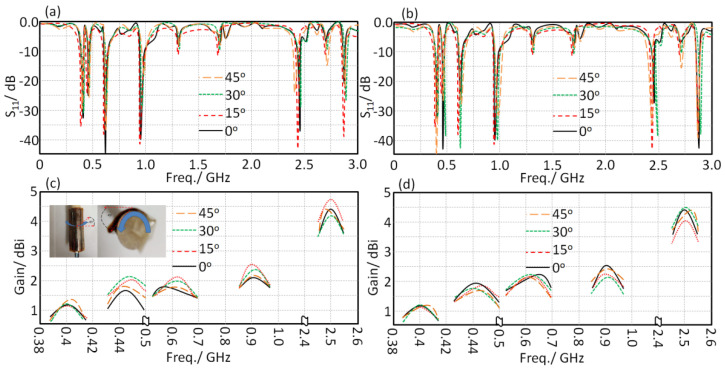
Antenna S_11_ and gain spectra measurements in free space and the wearable scenarios: (**a**) flat in free space, (**b**) flat on the human head, (**c**) bended in free space, and (**d**) bended on the human head.

**Figure 17 sensors-21-07960-f017:**
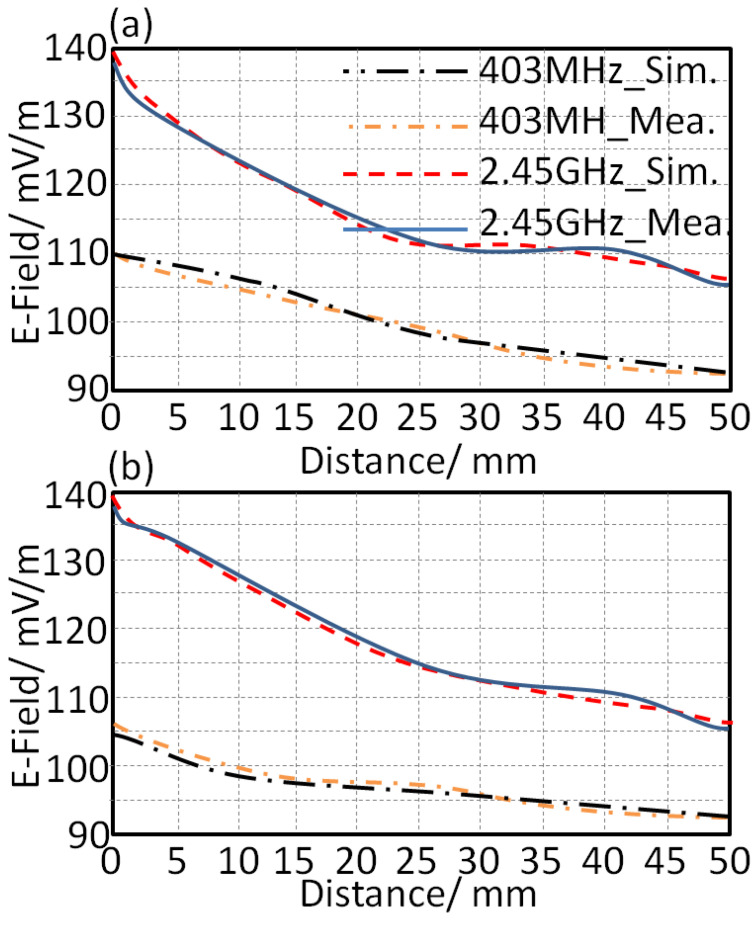
Antenna radiation leakage for the proposed profiles on the human head: (**a**) flat and (**b**) bended.

**Table 1 sensors-21-07960-t001:** Lumped element values of the equivalent circuit model in Figure 4.

Element	Value
RLH	12.2 Ω
RRH	50 Ω
GLH	0.1 S
GRH	4 S
CLH	1.1 pF
CRH	3.1 pF
LLH	3 nH
LRH	2.2 nH

**Table 2 sensors-21-07960-t002:** A comparison of the proposed antenna performance with respect to the published designs.

Refs.	Gain	Size	Center Frequency	Substrate Type
[1]	−20 dBi	λ/5	5 GHz	Fabric
[2]	3 dBi	λ/5	1.5 GHz	Solar panel polymer
[3]	4	0.12λ	2.45 GHz	Roger TMM10i
[4]	4.4	0.29λ	5.8 GHz	Unknown
[5]	2.33	3.27 mm	10.1, 24.6 GHz	Rogers RO3010
[6]	3.7	0.06 λ	2.45 GHz	Meta-cell
[7]	5.1	0.13 λ	2 GHz	Rogers RO4003C
[8]	7.1	16 mm	2.4, 5.8 GHz	Rogers 3210
The proposed work	1 dBi, 1.24 dBi, 1.48 dBi, 2.05 dBi, and 4.11 dBi	20 × 10 mm^2^	403 MHz, 433 MH, 611 Mz, 912 MHz, and 2.45 GHz	Polymer

**Table 3 sensors-21-07960-t003:** A comparison of SAR values.

References	Human Part	SAR (W/kg)	Centre Frequency (MHz)	Centre Frequency (MHz)
[5]	Head tissue	0.52~0.76	900–1800	900–1800
[6]	Head tissue	0.92	2400~2500 2500~2690	2400~2500
[7]	Head tissue		---	
[8]	Head tissue	0.45	900	---
[9]	Head tissue	0.45	850~2200	900
[10]	Wrist tissue	0.32~0.48	1960~1980	850~2200
[11]	---	0.28~0.43	930~1900	1960~1980
Proposed antenna	Head tissue	0.32~0.54	403, 2450	930~1900

## Data Availability

Not applicable.

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
