# Peer review of "A Flexible Metamaterial Based Printed Antenna for Wearable Biomedical Applications"

_sensors, 2021, doi:10.3390/s21237960_

Round 1

Reviewer 1 Report

This manuscript presents the A Flexible Metamaterial based Printed Antenna for Wearable Biomedical Applications. The work is interesting, but the paper is written and illustrated in a very casual way. Pictures and graphs quality is very poor. Please check the article throughout and make the corrections.  Some information's are missing from the manuscript side, which is very important to the readers. Some queries and suggestions are given in the following points for the betterment of the manuscript.

  1. Please include the findings or novelty of the proposed design in the abstract section to show the reader's particular contribution.
  2. Gain measurement should be included for all
  3. negative -εr should be in the proper format
  4. Figure 1 should be replaced by a good quality schematic diagram
  5. Graphs qualities are very bad, hard to understand. All Graphs should be improved.
  6. Figure 11 should be changed.
  7. Significant number of multiband MTM designs are available in the literature, so the authors need to specify the novelty of the proposed design. In this regard, it would be appropriate to present a comparative study of the proposed design with the existing designs in a tabular format. Need to discuss more previous article in the introduction section.
  8. As the antenna has been flexible in nature, please include the stability analysis of the antenna by studying the flexible characteristics in free space and the wearable scenario.
  9. Radiation pattern measurement in two frequencies (403MHz and 2.45GHz), Why?

Author Response

  • The paper is written and illustrated in a very casual way.

Answer: This is considered and thank you very much for the valuable comment.

  • Pictures and graphs quality is very poor.

Answer: This is considered in the revised version.

  • Please include the findings or novelty of the proposed design in the abstract section to show the reader's particular contribution.

Answer: This is considered in the revised version.

  • Gain measurement should be included for all frequencies

Answer: This is considered in the revised version.

  • negative -εr should be in the proper format.

Answer: This is fixed.

  • Figure 1 should be replaced by a good quality schematic diagram

Answer: This is considered in the revised version.

  • Graphs qualities are very bad, hard to understand. All Graphs should be improved.

Answer: This is considered in the revised version.

  • Figure 11 should be changed.

Answer: This is considered in the revised version.

  • Significant number of multiband MTM designs are available in the literature, so the authors need to specify the novelty of the proposed design. In this regard, it would be appropriate to present a comparative study of the proposed design with the existing designs in a tabular format. Need to discuss more previous article in the introduction section.

Answer: This is considered in the revised version.

  • As the antenna has been flexible in nature, please include the stability analysis of the antenna by studying the flexible characteristics in free space and the wearable scenario.

Answer: This is considered in the revised version.

  • Radiation pattern measurement in two frequencies (403MHz and 2.45GHz), Why?

Answer: This is considered in the revised version.

Reviewer 2 Report

The manuscript is devoted to a flexible printed multiband microstrip antenna based on metamaterials. In my opinion, the authors have obtained valuable simulation results experimentally confirmed with excellent agreement, worthy of publication. However, in its present form, the paper is written very carelessly and needs thorough revision before publication. I would like to stick to good patterns as far as the order of presented contents is concerned

Introduction: please reorder the content.
The part 'Operate the antenna at a good range and minimizing the dimension of wearable devices (...) complex designs, and low gain.' should be placed next, together with 'There are some tricky facing the production of wearable antennas (...)' because it concerns similar issues. In my opinion, the paragraph 'Day after day increased by the (...)' should be placed earlier, at the beginning of the discussion of antennas.

Antenna Geometry
Please describe precisely the geometry of the antenna and matching circuit, giving all dimensions and functions of the elements. Figure 1 is imprecise, please add a side view.

MTM Characterizations
Figure 2 does not include all dimensions of the unit cell, please add them. The sentence 'The authors treated this unit cell as explained later' is unfortunate, it does not explain anything.

Design Methodology
P
atch design: The authors mention antenna gain enhancement but don't support it with numbers and radiation characteristics - please add this information.

MTM Effects - please describe exactly what the parametric study was about, which dimensions were changed and to what extent, and comment on the results.

Radiation Leakage - for what input power level of the antenna were simulations and measurements performed?

Author Response

  • The paper is written very carelessly and needs thorough revision before publication.

Answer: This is revised acourding to your respected comments.

  • Introduction: please reorder the content. The part 'Operate the antenna at a good range and minimizing the dimension of wearable devices (...) complex designs, and low gain.' should be placed next, together with 'There are some tricky facing the production of wearable antennas (...)' because it concerns similar issues. In my opinion, the paragraph 'Day after day increased by the (...)' should be placed earlier, at the beginning of the discussion of antennas.

Answer: This is considered in the revised version.

  • Antenna Geometry: Please describe precisely the geometry of the antenna and matching circuit, giving all dimensions and functions of the elements. Figure 1 is imprecise, please add a side view.

Answer: This fixed in the revised version.

  • MTM Characterizations: Figure 2 does not include all dimensions of the unit cell, please add them. The sentence 'The authors treated this unit cell as explained later' is unfortunate, it does not explain anything.

Answer: This is considered in the revised version.

  • Design Methodology: Patch design: The authors mention antenna gain enhancement but don't support it with numbers and radiation characteristics - please add this information.

Answer: This is considered in the revised version.

  • MTM Effects: please describe exactly what the parametric study was about, which dimensions were changed and to what extent, and comment on the results.

Answer: This is considered in the revised version.

  • Radiation Leakage: for what input power level of the antenna were simulations and measurements performed?

Answer: This is considered in the revised version.

At the end, the authors thank the reviewers for their valuable comments. We wish to accept our paper in your respected journal.

Regards

Round 2

Reviewer 1 Report

Please discuss below points

  1. As the antenna has been flexible in nature, please include the stability analysis of the antenna by studying the flexible characteristics in free space and the wearable scenario.
  2. Gain measurement should be included for all frequencies (Gain vs frequency)

Author Response

  1. As the antenna has been flexible in nature, please include the stability analysis of the antenna by studying the flexible characteristics in free space and the wearable scenario.

Answer: This considered in the revised version in Fig. 16 for both S11 and gain spectra by bending the antenna from 0o to 45o on both free space and wearable scenarios.

  1. Gain measurement should be included for all frequencies (Gain vs frequency)

Answer: This is considered in the revised version.

Reviewer 2 Report

As the authors have taken my earlier suggestions into consideration, I have only a few less significant comments: 
1. the sections of the text added in the revised version need careful linguistic correction.
2. In my opinion, the abstract should be shorter, more concise. The last sentence of the abstract is unfinished (Finally, the proposed antenna measurements are compared to the simulated results to show good).
3. the authors indicate different novelties in the abstract than in the body of the article ('The novelty of the proposed design is basically by considering the complementary Minkowski geometry, therefore, such structure dominating the magnetic field instead of the electrical field. (...)') - please sort this out.
4. it seems to me that figure 2 still does not include all dimensions of EBG unit cell. Please correct the drawing and size it properly. 
5. section 5: 'In this section, the antenna radiations directions with respect to the human head are monitored at the frequency bands of interest' is written at the beginning. Meanwhile, Figure 9 analyzes the S-parameters. Please put this in order. 
6. there are two columns of SAR (W/kg) in Table 2 - what is the point?

Author Response

  1. As the antenna has been flexible in nature, please include the stability analysis of the antenna by studying the flexible characteristics in free space and the wearable scenario.

Answer: This considered in the revised version in Fig. 16 for both S11 and gain spectra by bending the antenna from 0o to 45o on both free space and wearable scenarios.

  1. Gain measurement should be included for all frequencies (Gain vs frequency)

Answer: This is considered in the revised version.

 Reviewer 2

  1. The sections of the text added in the revised version need careful linguistic correction.

Answer: Thank you very much for the kind advice. Therefore, the authors revised the entire version and fixed the language furthermore.

  1. In my opinion, the abstract should be shorter, more concise. The last sentence of the abstract is unfinished (Finally, the proposed antenna measurements are compared to the simulated results to show good).

Answer: This is considered in the revised version.

  1. The authors indicate different novelties in the abstract than in the body of the article ('The novelty of the proposed design is basically by considering the complementary Minkowski geometry, therefore, such structure dominating the magnetic field instead of the electrical field. (...)') - please sort this out.

Answer: This is considered in the revised version.

  1. It seems to me that figure 2 still does not include all dimensions of EBG unit cell. Please correct the drawing and size it properly. 

Answer: More geometrical details are included as well as the figure size is enlarged in t revised version.

  1. Section 5: 'In this section, the antenna radiations directions with respect to the human head are monitored at the frequency bands of interest' is written at the beginning. Meanwhile, Figure 9 analyzes the S-parameters. Please put this in order. 

Answer: This considered in the revised version.

  1. There are two columns of SAR (W/kg) in Table 2 - what is the point?

Answer: This is fixed.

At the end, the authors thank the reviewers for their valuable comments. We wish to accept our paper in your respected journal.

Regards

Round 3

Reviewer 1 Report

1.  Still not satisfying answer or explanation for this question, please go through some stability analysis related articles. (Q. As the antenna has been flexible in nature, please include the stability analysis of the antenna by studying the flexible characteristics in free space and the wearable scenario)

2. Please recheck the antenna gain measurement setup. Also, go through some articles related to frequency vs gain measurement. Please attach a picture of measurement. Also please explain how to control the bending during measurement when already attach to the human head (Figure 16).

Author Response

all answers in attached file

Round 4

Reviewer 1 Report

No comments!